# Observational Cross-Sectional Study on Mediterranean Diet and Sperm Parameters

**DOI:** 10.3390/nu15234989

**Published:** 2023-12-01

**Authors:** Gabriel Cosmin Petre, Francesco Francini-Pesenti, Andrea Di Nisio, Luca De Toni, Giuseppe Grande, Asia Mingardi, Arianna Cusmano, Paolo Spinella, Alberto Ferlin, Andrea Garolla

**Affiliations:** 1Unit of Andrology and Reproductive Medicine, Department of Medicine, University of Padova, 35128 Padova, Italy; gabriel.petre@rocketmail.com (G.C.P.); andrea.dinisio@unipd.it (A.D.N.); luca.detoni@unipd.it (L.D.T.); grandegius@gmail.com (G.G.); asia.mingardi@studenti.unipd.it (A.M.); andrea.garolla@unipd.it (A.G.); 2Clinical Nutrition Unit, Department of Medicine, University of Padova, 35128 Padova, Italy; francesco.francini@aopd.veneto.it (F.F.-P.); arianna.cusmano@studenti.unipd.it (A.C.); paolo.spinella@unipd.it (P.S.)

**Keywords:** Mediterranean diet, fertility, nutrition, sperm parameters, diet, infertility, foods, ultra-processed food

## Abstract

Infertility, affecting 15 to 25% of couples in the most developed countries, is recognized by the World Health Organization as a public health issue at a global level. Different causes are acknowledged to reduce fertility in both sexes. In particular, about 40–50% of cases recognize a male factor. Dietary habits and lifestyle are acknowledged to influence sperm quality and are therefore important modifiable factors in male reproductive health. Conditions such as overweight/obesity, impaired glucose metabolism and determinants of metabolic syndrome, together with unhealthy lifestyle behavior, i.e., smoking cigarettes and physical inactivity, are suggested to have a negative impact on male fertility. While individual elements and characteristics of the Western diet and habits are considered risk factors for male infertility, the Mediterranean diet (MD) seems to promote reproductive potential for improving sperm quality. It is also interesting to note that previous observational studies reported a positive correlation between the consumption of the single food classes of the MD pattern (i.e., vegetables and fruits, poultry, fish and seafood, whole grains, low-fat dairy products) and the quality of several sperm parameters. To evaluate the relationship between sperm parameters and MD adherence, we performed a cross-sectional study on the seminal data of 300 males (mean age 34.6 ± 9.1 years) who spontaneously referred to our center of reproductive medicine. The evaluation of adherence to MD was performed with a validated 14-point Mediterranean Diet Adherence Screener (MEDAS) questionnaire. Our findings showed that sperm parameters such as sperm count, motility, viability and normal morphology are significantly and positively correlated with MEDAS, independently of BMI and age. In addition, the application of an ROC curve on MEDAS value vs. seminal alterations identified 6.25 as the score threshold value below which altered sperm parameters were more likely to occur [AUC = 0.096 (CI: 0.059–0.133; *p* < 0.00)]. Therefore, adhering to the MD with at least a MEDAS score of 6.26 increases the probability of normozoospermia. Moreover, subjects who had a MEDAS value lower than 6.25 had an Odds Ratio of 6.28 (CI = 3.967–9.945) for having at least one altered sperm parameter compared to those who were more adherent to the MD. In conclusion, our findings show that a higher adherence to the MD is associated with better semen parameters, in particular in relation to sperm count, sperm concentration, typical sperm morphology, and sperm progressive motility.

## 1. Introduction

Nowadays, infertility affects 15 to 25% of couples in Western countries. Reduced fertility can have different causes and affects both sexes [1]. In particular, about 40–50% of cases are ascribable to male factors. In addition to congenital problems, certain conditions such as overweight/obesity, diabetes, insulin resistance, atherosclerosis, and hypertension, together with unhealthy lifestyle behavior such as cigarette smoking, alcohol abuse, and physical inactivity, have been suggest to have a negative impact on male fertility [2,3]. Moreover, there is growing awareness of the relationship between unhealthy diet and male infertility [4].

Currently, there is a large body of research supporting the hypothesis that some foods may have beneficial effects on semen quality (fish, seafood, poultry, whole grains, fruits and vegetables, and low-fat dairy products) or, on the contrary, may be harmful (i.e., preserved and processed meat, foods high in saturated fatty acids, alcohol, and sugary beverages and sweets) [5]. In relation to individual nutrients, a low consumption of saturated fatty acids and trans fatty acids and an adequate intake of some nutrients such as omega-3 fatty acids, antioxidant molecules, and vitamins are associated with greater reproductive health [6,7,8]. In fact, it has been shown that a higher dietary intake of vitamins, various polyphenols, and carotenoids was positively correlated with better sperm quality [9]. On the other hand, dietary patterns are clearly associated with sperm quality, suggesting that nutritional interventions could play a key role in male fertility preservation, as opposed to the inadequate eating habits of the Western diet [10]. The Western diet is characterized by a high consumption of ultra-processed foods with a high intake of sugars and fats, inducing a nutritional imbalance due to the introduction of excessive calories. Hyperinsulinemia and hyperglycemia, which are the hallmarks of insulin resistance and typical of obesity, have been reported to be responsible for sperm’s altered glucose metabolism, thus playing a role in impaired glycolysis and consequently in increased oxidative stress [11]. In addition to hyperinsulinemia, obesity is characterized by hyperleptinemia. High levels of these hormones have been shown in the seminal plasma of dysmetabolic subjects and associated with an impairment of sperm quality and reproductive function [12].

The Mediterranean diet (MD), more correctly known as the Mediterranean food model, is a modern nutritional recommendation inspired by the food traditions of countries of the Mediterranean Sea basin. The importance and effectiveness of MD in the prevention of non-communicable diseases, especially those affecting the cardiovascular system, has been highlighted by many observational and experimental studies [13]. This diet model is characterized by a high consumption of seasonal fresh fruits and vegetables, cereals, nuts, and legumes, a moderate consumption of fish and wine, and a very low consumption of dairy products and meat, where olive oil is the main source of fat. Thus, the Mediterranean diet is rich in fibers, monounsaturated fatty acids, and antioxidants and low in saturated fat [14]. Seasonal fruits and vegetables are rich in antioxidant and anti-inflammatory molecules, which can act as regulators of the redox status both by acting as a Reactive Oxygen Species (ROS) scavenger and as gene modulators [15]. These natural dietary compounds have been suggested to ameliorate sperm parameters and even to improve DNA fragmentation and chromatin integrity [16] and increase sperm concentration, sperm motility, and sperm viability [17], variables that are closely linked to the sperm quality. It is also interesting to highlight that previous observational studies have reported a positive correlation between the consumption of individual classes of healthy foods and the quality of several sperm parameters.

Since MD has been shown to have beneficial effects in several intermediate metabolic outcomes, such as inflammation and oxidative stress, we can hypothesize that a higher MD adherence may have benefits in terms of reproductive health and sperm parameters, as highlighted in other previous papers in both interventional [18] and observational studies [19,20,21,22].

In the present observational study, we investigate the relationship between the level of adherence to the MD model and sperm parameters in a large cohort subject undergoing semen analyses, taking into account many confounding factors which, in our opinion, in previous publications on the same topic, could have been managed better.

## 2. Materials and Methods

### 2.1. Study Population and Nutritional Habits

This is an observational cross-sectional study aiming to evaluate the relationship between sperm parameters and MD adherence in subjects who consecutively referred to the Andrology and Reproductive Medicine Unit, University Hospital of Padova, for semen analysis. All subjects who voluntary agreed to participate in the study underwent a medical history and an anthropometric evaluation of body weight and height and a lifestyle interview including a dietary questionnaire and smoking habits. Body mass index (BMI) was calculated as the ratio between body weight (in kg) and the square of the height (in meters). The MD adherence was assessed through the same validated 14-point a priori Mediterranean Diet Adherence Screener (MEDAS), under the supervision of the same dietitian. The MEDAS questionnaire provides three levels of adherence (≤5 low adherence, 6–9 medium adherence, ≥10 high adherence) [23] (see Appendix A). Included subjects also underwent semen collection by masturbation into sterile containers after 2–7 days of sexual abstinence. Samples were allowed to liquefy for 30 min and were examined for semen pH, semen volume (mL), sperm concentration (10^6^ cells/mL), total sperm count (10^6^ cells/ejaculate), sperm viability (%), sperm progressive motility (%), and typical sperm morphology (%) according to the WHO 6th edition manual 2021 [24]. Moreover, semen parameters thresholds, allowing characterization as normozoospermia, oligozoospermia, and asthenozoospermia, were based on the 6th percentiles from WHO criteria.

### 2.2. Ethical Approval

The study was approved by the Ethics Committee of the University Hospital of Padova (protocol no. AOP3205, granted on 25 October 2023), and each participant signed informed consent. The study was conducted in accordance with the principles expressed in the Declaration of Helsinki.

### 2.3. Inclusion and Exclusion Criteria

All subjects aged 18 to 45 years old who consecutively referred to the Andrology and Reproductive Medicine laboratory between September 2022 and July 2023 were included in the study.

We excluded subjects with a current controlled diet or who had carried one out in the last 3 months, high fever (≥38 °C for at least two days) in the last 3 months, varicocele, metabolic syndrome, malignancies, history of cryptorchidism, history of testicular cancer, endocrinopathies, semen infections, genetic causes of infertility, and the use of medical treatments or dietary supplements in the 3 months preceding the study, and patients that on the day of semen collection did not have the required sexual abstinence. Furthermore, physical activity may affect male fertility, although data from the scientific literature are conflicting [25]. For this reason, we excluded from this study subjects who reported undertaking physical activity for more than 3 h per week.

### 2.4. Statistics

All statistical analyses were performed using SPSS software (Version 25, SPSS Inc., Chicago, IL, USA). Values of *p* < 0.05 were considered statistically significant. The results were expressed as mean ± standard deviation (SD). The differences between continuous variables were analyzed with ANOVA. The differences between discrete variables were analyzed with a Chi-square test or Fisher test (if the expected count was <5). The Pearson correlation index or the Spearman correlation index for non-normally distributed variables were used to describe the correlations between variables. On the basis of correlation analyses, we then performed multilinear regression analyses to compare the semen parameters most associated with MEDAS value, after adjusting for the following confounders: age, BMI, and smoking habits.

The inclusion and exclusion criteria were defined for *p* < 0.05 and *p* > 0.1, respectively. Ordinal regression analyses were performed to assess the impact of MEDAS value on the increase in the number of semen alterations. The Jonckheere–Terpstra test was performed to calculate the *p* for the trend in the association between the number of seminal changes and variables of interest.

An ROC curve was calculated by plotting sensitivity values on the y-axis and 1-sensitivity values on the x-axis. The area under the curve (AUC) was determined and evaluated using the Sweets classification and the cutoff values for each considered variable were calculated with Youden’s S statistics.

## 3. Results

### 3.1. Population Characteristics

In the period 22 September–23 July, 300 consecutive males were enrolled in this study. Their demographic and clinical data are reported in Table 1. The mean age was 34.6 ± 4.1 SD and BMI was 24.3 ± 4.1 SD (a young and normal weight sample). The mean MEDAS value of the whole sample was 7.6, indicating a medium adherence to MD. In particular, 32.3% of subjects had a low MD adherence, 36.7% had a medium MD adherence, and 31.0% had a high MD adherence. Moreover, 213 (71%) subjects were non-smokers and 87 (29%) were smokers. It is also interesting to note that the two groups had a different MD adherence: the mean MEDAS of non-smokers was 7.87 ± 2.98 SD, and the value for smokers was 7.01 ± 2.85 SD. The difference between the two groups was significant (*p* = 0.022).

### 3.2. Correlation Analyses

In order to evaluate the possible role of MD adherence on sperm parameters, a correlation analysis was performed between the MEDAS value and sperm parameters. All sperm parameters, with the exception of pH (*p* = 0.534), were positively and significantly associated with MEDAS. In particular, sperm concentration, total count, progressive motility, viability, and typical sperm morphology were positively and strongly associated with the MEDAS value. Consequentially, the percentage of not-motile spermatozoa showed a negative correlation (r = −0.498; *p* < 0.001) with MEDAS. Of note, the strongest level of association observed was between MEDAS and sperm count, then typical sperm morphology, followed by sperm progressive motility. No significant correlation was found between MEDAS and BMI or age (*p* = 0.093 and *p* = 0.737, respectively) (Figure 1a–j).

Regarding smoking habit, a correlation analysis was also performed between MEDAS value and sperm parameters, adjusted for age, BMI, and smoking. All sperm parameters, with the exception of pH (*p* = 0.538), were still positively and significantly associated with MD adherence (see Appendix A).

After grouping subjects according to MD adherence (low, medium, and high) the multivariate analysis showed a significant difference in sperm parameters in the three groups (Figure 2). The characteristics of sperm concentration, count, semen volume, progressive motility, sperm viability, and morphology were significantly higher in subjects reporting a medium and high adherence to MD. In particular, sperm count and typical morphology were, respectively, 38.67 (10^6^ cells); 3.39 (%) for low, 102.03 (10^6^ cells); 5.12 (%) for medium, and 179.29 (10^6^ cells); 6.98 (%) for high MD adherence.

Considering sperm parameters as categorical variables based on the sixth percentiles according to the WHO manual, we performed a logistic regression analysis to evaluate the presence and number of sperm alterations in relation to MEDAS (Figure 3). In particular, altered sperm parameters were considered: semen pH < 7.2, semen volume < 1.4 (mL), sperm concentration < 16 (10^6^ cells/mL), total sperm count < 39 (10^6^ cells/ejaculate), sperm viability < 54 (%), sperm progressive motility < 30 (%), and typical sperm morphology < 4 (%) [26].

Using this analysis, we observed that subjects with normozoospermia had the highest MEDAS values. The cumulative number of sperm alterations was inversely related to the MEDAS value (*p* for trend < 0.001).

Moreover, the prevalence analyses of at least one semen alteration in any of the considered sperm parameters (under the sixth percentile of WHO criteria) in each MEDAS category showed that 92.5% (99 out of 107) of the subjects with high adherence were normozoospermic (*p* < 0.001) (Table 2). The prevalence of normozoospermia progressively decreased to 57.5% and 8.80% in, respectively, the medium and low adherence groups. Interestingly, patients in the lower MD adherence group showed at least one sperm alteration in more than 90% of cases.

Finally, an ROC curve was performed in order to identify whether the MEDAS value was able to predict normal sperm parameters (Figure 4). The MEDAS threshold that best identified subjects with at least one altered sperm parameter was 6.25. A MEDAS value of 6.26 predicted normozoospermia, with an AUC = 0.096 (CI: 0.059–0.133; *p* < 0.00).

Interestingly, using the 6.25 cut-off of the ROC curve, in order to obtain a binary score of MEDAS, the odds ratio of having at least one altered sperm parameter was 6.28 (CI = 3.967–9.945; *p* < 0.001) for the reduced MEDAS score, compared to being more adherent to the Mediterranean dietary model.

## 4. Discussion

In this study, we provide evidence that the degree of adherence to the Mediterranean diet, evaluated through the MEDAS questionnaire, significantly correlates with sperm parameters and predicts the occurrence of their alteration independently of age and BMI, smoke, and physical activity.

Fertility, and in particular male fertility, is acknowledged to be affected by several environmental factors and lifestyle habits, including smoking, pollution, alcohol consumption, low physical activity and, last but not least, dietary regimen [27,28]. In this regard, a hypercaloric diet and weight gain may exert direct noxious effects on male reproduction by favoring local oxidative stress [29]. It is acknowledged that overweight and obesity negatively interfere with the hypothalamic–pituitary–gonadal axis, and the resulting hypogonadism affects both spermatogenesis and energy metabolism in a vicious circle [28,30]. However, aside from this clear two-way negative cross-talk between endocrine alterations and metabolic disorders, there is a considerable “gray zone” in which this association appears much less clear. This is the case for the so-called metabolically unhealthy normal weight phenotype, accounting for up to 30% of normal weight subjects, in which oligozoospermia represents a common feature [31,32]. In this frame, the adherence to a healthy dietary pattern such as MD appears to be a useful intervention to prevent a metabolically unfavorable phenotype in normal-weight men. Recently, Golzarand et al., in a study evaluating 1299 subjects, showed that the adoption of MD or the Dietary Approaches to Stop Hypertension-diet, largely overlapping with MD, was, respectively, associated with a 46% and 47% lower risk of gaining a metabolically unhealthy phenotype, highlighting the driving force of MD as a nutritional approach [33].

The MD model is believed to be positively associated with male fertility status, thanks to its low level of saturated fatty acids, trans fatty acids, adequate levels of monounsaturated fatty acids, and polyunsaturated fatty acids fraction (PUFA), such as omega-3 and omega-6 [34]. Furthermore, this nutritional model provides excellent levels of antioxidant molecules, minerals, and vitamins. By supplying adequate quantities and qualities of fatty acids with the diet, especially in relation to the omega-3/omega-6 ratio, optimal sperm maturation can be achieved. Indeed, dietary fatty acids are constituents of the sperm cell membrane [35]. A diet poor in saturated fatty acids and cholesterol, with a low PUFA ratio, seems to improve semen quality [36,37], ameliorating membrane fluidity [38], reducing oxidative stress, and increasing mitochondrial function [39].

It is also interesting to point out that previous observational studies reported a positive correlation between the consumption of the single food classes of the MD pattern (i.e., vegetables and fruits, poultry, fish and seafood, whole grains, low-fat dairy products) and the quality of several sperm parameters [40,41]. Moreover, a systematic review and meta-analysis of observational studies, evaluating the association between some dietary patterns and semen quality, concluded that a healthy eating pattern could have a positive impact only on semen concentration, but not on other parameters [42]. Is important to underline that the analyzed sample was small and the included studies were very heterogeneous. Instead, Salas-Huetos and colleagues, in a large systematic review on observational studies, found that diets rich in foods classes such as fish, poultry, cereals, vegetables, and fruits, and low in saturated and trans fatty acids, were positively associated with improvements in several sperm parameters [43].

Therefore, nutrition is recognized to affect sperm quality through several mechanisms. As an example, the supply of essential nutrients such as zinc, selenium, essential fatty acids, folic acid, and vitamin B12 is recognized to be important in the maturation of spermatozoa [44]. In addition, weight status, the influence of physiological mechanisms involved in spermatogenesis (hormonal modulation and red-ox balance), and the physiology of the accessory glands, are strongly linked to sperm quality [45,46]. A common denominator of all these aspects is chronic pro-inflammatory conditions, both systemic and testicular, a complex network that could be modulated through nutrition [47,48]. In this context, inflammation may also affect male reproductive potential through anatomical or functional changes in the accessory gland [48]. In fact, it is believed that eating habits are a part of various prostate diseases, from benign prostatic hyperplasia (BPH) to cancer [49]. Diet and certain nutrients can favorably influence prostate health, reducing inflammatory processes and consequently potentially influencing sperm parameters [49]. Evidence shows that prostate concerns are less frequent in men with better adherence to a healthy diet. Moreover, lycopene, zinc, vitamin D, phytosterols, and omega 3 fatty acids may reduce the risk of symptomatic BPH [50].

In the present study, we evaluated semen parameters in 300 non-obese males, in relation to the individual adherence to MD, evaluated through the MEDAS score. Interestingly, all major seminal predictors of fertility potential, including sperm count, sperm motility, typical sperm morphology, and sperm viability, were significantly and positively correlated with the MEDAS score. This evidence suggests the role of the nutritional regimen as a possible marker of sperm abnormalities in those patients who, otherwise, remain defined as idiopathic. Those patients would be susceptible to an in-depth analysis in search of possible dysmetabolism. Our data are consistent with a growing number of reports on this topic. Karayiannis et al. evaluated 225 men from couples attending a fertility clinic [19]. Men in the lowest tertile of MD adherence score showed worse sperm count, total motility, and concentration values, whilst a high-MD adherence score was significantly associated with higher sperm quality. Ricci et al. evaluated the MD adherence in 309 male partners of sub-fertile couples undergoing assisted reproduction techniques [20]. In this study, the MD regimen was positively associated with sperm concentration and count, but not with seminal volume. Importantly, neither of the two previous studies distinguished participants according to congenital fertility problems, thus introducing a bias in the evaluation of the diet role. Salas-Huetos et al. evaluated 106 healthy young participants, concluding that an adherence to MD was positively associated only with sperm motility [21]. Lastly, Cutillas-Tolin et al., in a study on 215 healthy male college students, evidenced that higher MD adherence was positively associated only with sperm count [22].

Compared to previous studies, our data are particularly strengthened by the considerable sample size, the homogeneous evaluation of the dietary regime carried out by a single dietitian, and the exclusion of known andrological, lifestyle, and metabolic causes of infertility such as andrological problems, smoking habits, physical activity, overweight or obesity, and higher age, supporting an independent role of the dietary regimen. Nonetheless, we acknowledge a major limitation of the present study, which is its observational and cross-sectional nature. In fact, a cross-sectional design cannot determine whether MD adherence and semen parameters are causally related. Thus, although we excluded several potential confounding factors, these results should be considered with caution because of the many factors that can affect semen quality, and sperm parameters cannot be directly translated into fertility terms. Further well-designed prospective observational studies and clinical trials on the current topic are therefore recommended.

## 5. Conclusions

In conclusion, our findings confirm a better semen quality in males with higher adherence to the MD. On the contrary, when adherence decreased, the number of sperm alterations increased. In fact, in our sample, subjects who were more adherent to MD (even medium adherence, a MEDAS value > 6.25) had an approximately six times less probability to have sperm alterations compared to those who were poorly adherent. In particular, grouping patients by MEDAS value, we observed that greater adherence to MD was positively correlated with sperm concentration, sperm count, sperm progressive motility, semen volume, sperm viability, and typical sperm morphology independent of age, BMI, and smoking habits.

This dietary food model is rich in compounds that can be useful to sperm composition and function, suggesting a physiological favorable role in male fertility. In contrast, poor diet quality, even when not necessarily characterized by overweight, is more frequently associated with alterations in sperm parameters and male infertility. Therefore, nutritional counseling appears to be a readily implementable, non-invasive clinical practice advisable in all men with signs of alteration in sperm quality.

## Figures and Tables

**Figure 1 nutrients-15-04989-f001:**
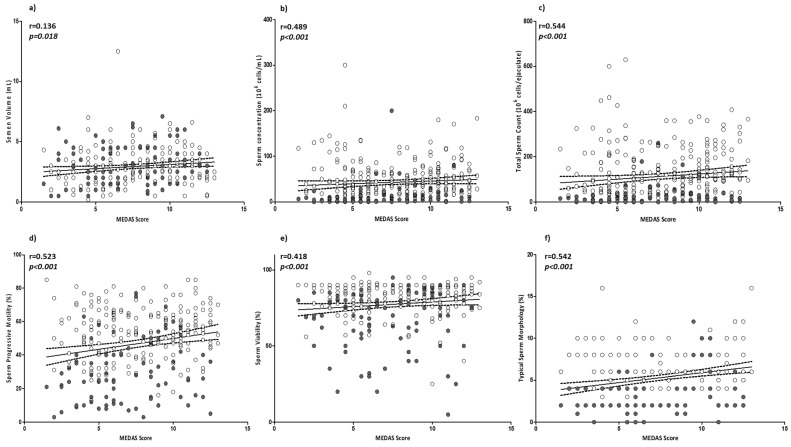
Correlation analyses between MEDAS and sperm parameters: (**a**) semen volume; (**b**) sperm concentration; (**c**) total sperm count; (**d**) sperm progressive motility; (**e**) sperm viability; (**f**) sperm typical morphology; (**g**) semen pH; (**h**) non-motile sperm; (**i**) BMI; and (**j**) age. Filled circles = smokers; empty circles = non-smokers. The 95% confidence intervals are represented by dotted lines. Significant *p* values are highlighted in cursive.

**Figure 2 nutrients-15-04989-f002:**
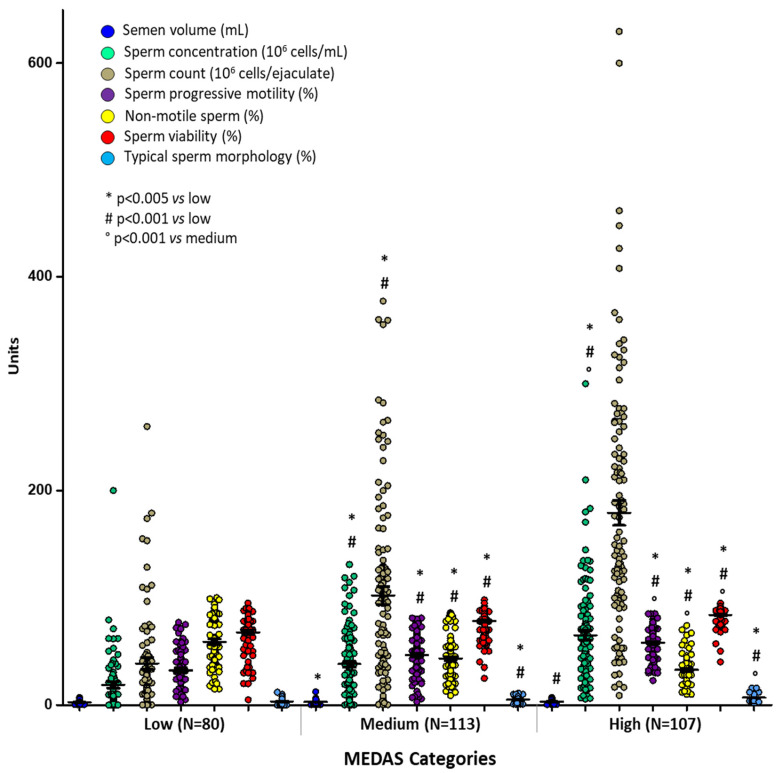
Means ± SD of sperm parameters across MEDAS categories. *p* values were calculated by multivariate analyses corrected for age and BMI. * *p* < 0.005 vs. low; # *p* = 0.001 vs. low; ° *p* = 0.04 vs. medium.

**Figure 3 nutrients-15-04989-f003:**
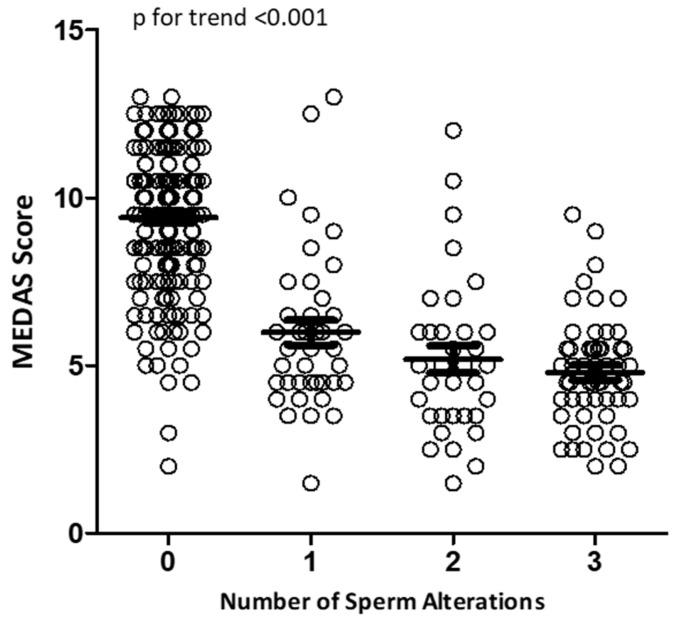
Logistic regression analysis of MEDAS values in relation to the number of sperm alterations.

**Figure 4 nutrients-15-04989-f004:**
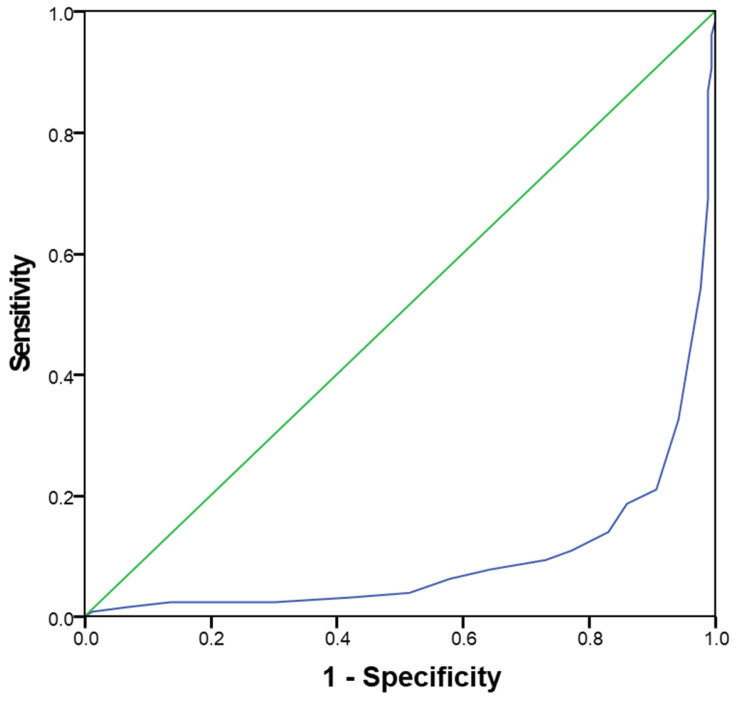
ROC curve for MEDAS value able to identify subjects with normozoospermia.

**Table 1 nutrients-15-04989-t001:** Sample characteristics. Data are presented as mean ± SD.

Variables	Mean		SD
Age (years)	34.6	±	9.1
BMI (Kg/m^2^)	24.3	±	4.1
MEDAS value	7.6	±	2.9
Semen Volume (mL)	2.94	±	1.51
Semen pH	7.53	±	0.23
Sperm Concentration (10^6^ cells/mL)	42.54	±	41.39
Total Sperm Count (10^6^ cells/ejaculate)	112.69	±	109.29
Progressive Motility (%)	46.79	±	19.55
Non-motile Sperm (%)	43.67	±	20.54
Sperm Viability (%)	77.44	±	15.49
Typical Sperm Morphology (%)	5.32	±	2.81

**Table 2 nutrients-15-04989-t002:** Chi-square test; MEDAS categories vs. number of sperm alterations.

			Number of Alterations			
			0	1	2	3 or more
MEDAS Categories	Low	Subjects	7	18	20	35
		% within MEDAS	8.80%	22.50%	25.00%	43.80%
	Medium	Subjects	65	18	11	19
		% within MEDAS	57.50%	15.90%	9.70%	16.80%
	High	Subjects	99	4	3	1
		% within MEDAS	92.50%	3.70%	2.80%	0.90%
*p* < 0.001						

## Data Availability

The data presented in this study are available on request from the corresponding author.

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
