# Peer review of "Observational Cross-Sectional Study on Mediterranean Diet and Sperm Parameters"

_nutrients, 2023, doi:10.3390/nu15234989_

Round 1
Reviewer 1 Report
Comments and Suggestions for Authors
To my opinion, the use of the term "large" in the title, is an exaggeration. With 300 men participated, a study like this is not large; it is of medium size.
line 79 "height and high": please delete "and high"
line 91 "All subjects aged to 18 from 45": please rephrase "All subjects aged from 18 to 45"
line 186 "on the male reproductive"" please change "on the male reproduction"
Comments on the Quality of English LanguagePlease, check the English language
Author Response
We thank the Reviewer for her/his positive comments, we have made all the changes to the manuscript in accordance with what was suggested
Reviewer 2 Report
Comments and Suggestions for Authors
Gabriel Cosmin Petre, et al. investigated 300 subjects and discovered that the Mediterranean Diet (MD) could significantly improve the semen quality including normal sperm morphology, progressive motility, sperm count and etc. Actually, the type of this cross-sectional study relevant to MD and semen quality was already published by many researchers, so that the current study is devoid of innovation, although they emphasized that the presented manuscript contained more information or clinical data compared to other already published articles. On the other hand, the subjects who enrolled into the study did not had any information about his physical state including physical activity, daily MD food quantity, and etc., which may also influence the sperm quality. Accordingly, the authors need to redesign the questionnaire which including more valuable information and improve the manuscript indeed.
Comments on the Quality of English LanguageNo comments
Author Response
We thank the Reviewer for her/his positive and precious comment. In relation to physical activity, we specified in the materials and methods that we excluded from our case studies those who performed more than three hours a week of physical exercise, since, as is rightly underlined, this can influence the sperm quality. However, regarding the used questionnaire, as we have specified, it is a previously validated questionnaire, therefore we did not design it. The MEDAS questionnaire is a semi-quantitative, it refers to weekly food frequencies consumption (FFQ) from which we cannot deduce the quantity of the various foods consumed or any other nutritional information.
Reviewer 3 Report
Comments and Suggestions for Authors
This is a good publication but presenting only a confirmatory study with a limited insight, because the effect of MD on sperm parameters has been revealed previously. Moreover, the manuscript suffers from lower clarity and information content.
Although pinpointing the importance of smoking habits for sperm quality, the study does not even reveal the number of smokers involved and their proportions in the MD categories. The way of excluding the effect of smoking is insufficiently described- a graph should be given (as supplement?). The same applies for age and BMI.
There are two Figures no.3, which must be resolved.
Information contents of Fig.1 should be increased by highlighting age/smoking status of the participants by different colors and/or point shapes. 95% confidence intervals must be shown for all regression lines in Fig.1. The exact p values have to be shown (not just “p<0.001”).
Bar graphs are no longer being used for their low information content - data distribution cannot be seen (https://doi.org/10.1371/journal.pbio.1002128). Thus bar graphs in Figs 2 and 3 must be changed to violin plots or sina plots with sample sizes indicated.
Table 2: The “Number of sperm alterations” must be explained in DETAIL (the reference to WHO is necessary but not sufficient - details are necessary, because the readers do not want to read the WHO manual).
Unless publishing all data points of every participant, ALL possible graphs should be included (some can be in a supplement), incl. the effect of age, BMI, smoking on sperm parameters, as well as for the analyses mentioned on lines 138, 140-141.
Lines 209-220: this literature [22-25] should be also mentioned in the Introduction. What new insight does bring the current study? A table would be useful.
While the paragraph on lines 221-232 correctly describes the limitation of the study, this is somehow denied in the concluding sentences (lines 233-239), where the authors should make much more clear, where do speculations start from or provide data/references for these claims.
The MEDAS questionnaire used should be disclosed as Supplementary info. Did all participants obtained exactly the same one? If not, the versions should be indicated and number of participants disclosed for each version. If the questionnaire does not reveal the details of MD, these details must be described.
Lines 28-30 (“In addition…”): this sentence is too long and cannot be understand by general reader, thus it should be removed or rephrased.
Comments on the Quality of English LanguageMinor comments:
Line 24: “±9,1“ must be “±9.1“
Author Response
This is a good publication but presenting only a confirmatory study with a limited insight, because the effect of MD on sperm parameters has been revealed previously. Moreover, the manuscript suffers from lower clarity and information content. Although pinpointing the importance of smoking habits for sperm quality, the study does not even reveal the number of smokers involved and their proportions in the MD categories. The way of excluding the effect of smoking is insufficiently described- a graph should be given (as supplement?). The same applies for age and BMI.
We thank the Reviewer for her/his positive comments. In agreement with your precious indication, we implemented the text (result part) with the smoke information. Moreover, we also made a table (included in the supplements) with data relating to correlations between MEDAS value and semen parameters adjusted for BMI, age and smoking
There are two Figures no.3, which must be resolved.
Thanks for this indication that was resolved in the new manuscript
Information contents of Fig.1 should be increased by highlighting age/smoking status of the participants by different colors and/or point shapes. 95% confidence intervals must be shown for all regression lines in Fig.1. The exact p values have to be shown (not just “p<0.001”).
In accordance with these very useful suggestions, we have redone the graphs showed in Figure 1 the required data. Regarding the significance value, the graphical output of SPSS is with only 3 values ​​after the point "p = .001"
Bar graphs are no longer being used for their low information content - data distribution cannot be seen (https://doi.org/10.1371/journal.pbio.1002128). Thus bar graphs in Figs 2 and 3 must be changed to violin plots or sina plots with sample sizes indicated.
In accordance with these very useful graphical suggestions, we changed the bar graphs with sina plots.
Table 2: The “Number of sperm alterations” must be explained in DETAIL (the reference to WHO is necessary but not sufficient - details are necessary, because the readers do not want to read the WHO manual).
In relation to this question, the data in table 2 refers to any alteration of the sperm parameters, it does not mean a specific alteration. Therefore, it is the same values ​​present in table 1 that present an altered value (below the 5th percentile according to the WHO criteria). We have improved the description in the text adding WHO 5th percentile reference values.
Unless publishing all data points of every participant, ALL possible graphs should be included (some can be in a supplement), incl. the effect of age, BMI, smoking on sperm parameters, as well as for the analyses mentioned on lines 138, 140-141.
As per the previous answer, we redid the graphs in figure 1 implementing them with information on BMI, age in relation to the correlation with the MEDAS data
Lines 209-220: this literature [22-25] should be also mentioned in the Introduction. What new insight does bring the current study? A table would be useful.
We have explained this information among the strengths and weaknesses of our work (line 317 to 328), and we cited literatur [22-25] in the introduction
While the paragraph on lines 221-232 correctly describes the limitation of the study, this is somehow denied in the concluding sentences (lines 233-239), where the authors should make much more clear, where do speculations start from or provide data/references for these claims. The MEDAS questionnaire used should be disclosed as Supplementary info. Did all participants obtained exactly the same one? If not, the versions should be indicated and number of participants disclosed for each version. If the questionnaire does not reveal the details of MD, these details must be described.
We have implemented this important information in the text and adding MEDAS questionnaire as a supplementary material, thanks for the useful suggestion. Moreover we have changed the sentences regarding "speculation".
Lines 28-30 (“In addition…”): this sentence is too long and cannot be understand by general reader, thus it should be removed or rephrased.
This phrase was changed accordingly
Comments on the Quality of English Language
Minor comments:
Line 24: “±9,1“ must be “±9.1“
Round 2
Reviewer 2 Report
Comments and Suggestions for Authors
It is OK now except for some errors correction.
Comments on the Quality of English LanguageIt is OK now except for some errors correction.
Author Response
Dear reviewer, thank you for your support. We have revised the manuscript as your suggestion regarding the English language.
Reviewer 3 Report
Comments and Suggestions for Authors
The authors improved the presentation and clarity of the manuscript, but
the article does not reveal anything new. I may be wrong, but I have not found a study where the diet effect on sperm would be separately investigated in smokers, is it possible to address this question in this subset of 87 smokers? Given the large effect of smoking on sperm quality, known from previous studies and apparent in the presented data, does adherence to MD display a similar trend in smokers as in the non-smokers?
The descriptions of the significance of MEDAS value 6.25 in the Abstract is still incomprehensible to anyone but the authors, because no guide of scoring the supplementary questionaire is provided. For the same reason, the study is not reproducible, thus the exact and detailed description how was the questionaire employed to get to the MEDAS value has to be given.
In the Discussion, the authors fail to mention that another limitation of the study is that the sperm parameters cannot be directly translated into the terms of sperm function and fertility, so it is unclear, whether the improvement of sperm parameters presented here has any practical effect (actual effect on fertility).
Line 96 "allowing", not "alloweing".
Author Response
Dear reviewer, thank you for your further support in improving our manuscript. As your precious suggestions, we have implemented in the text the data relating to the MEDAS value between smokers and non-smokers; there is a difference (about 1 point) and this is significant. Furthermore, there are no other significant differences in semen parameters between smokers and non-smokers. We have implemented the description of the MEDAS value of 6.25 in the Abstract. We have modified the presentation of the MEDAS questionnaire in the supplementary material to make it more understandable. Finally, we have included the detail you suggested in discussion regarding the fact that the sperm parameters do not necessarily reflect fertility.